# Solitons in Weakly Non-linear Topological Systems: Linearization, Equivariant Cohomology and K-theory

D. Sheinbaum[1]

**1** Departamento de Física, Centro de Investigación y de Estudios Avanzados del Instituto Politécnico Nacional, P.O. Box 14-740, CP. 07000, México D.F., Mexico.
* daniel.sheinbaum@cinvestav.unam.mx

January 4, 2023

## Abstract

There is a lack of knowledge about the topological invariants of non-linear $d$-dimensional systems with a periodic potential. We study these systems through a classification of the linearized NLS/GP equation around their soliton solutions. Stability conditions under linearized (mode) adiabatic evolution can be interpreted topologically and we can use equivariant cohomology for their classification. We further construct purely non-linear global invariants using the space of symmetry-breaking soliton solutions $\mathcal{M}$, given by $K^{-1}(\mathcal{M})$. We conjecture that these new phases signal a global bifurcation for the non-linear system and discuss their physical interpretation.

# 1   Introduction

The field of topological systems in photonics [1], exciton-polaritons [2] and Bose-Einstein Condensates (BEC) [3] has brought to the fore the interplay between non-linear effects such as solitons [4] and linear topological phases of matter [5], [6], [7]. An interesting question arises when we couple a non-linear medium to a periodic potential [8], particularly when the linear system is topologically non-trivial, e.g. quantum Hall effect (QHE), SQHE and topological insulators (TI) [1]. Are systems which combine both still topological? And if so, which topological invariants classify these new non-linear systems? Though there are already many theoretical and experimental results on the stability of solitons, non-linear Bloch waves and their interactions, as well as resulting macroscopic properties of materials [2], [9], [10], there are not many proposals for topological invariants classifying these systems (see [11], [12] for some results in this direction). For weakly non-linear systems, the Non-linear Schrödinger/Gross-Pitaevskii equation (NLS/GP) [4], [9], [13] and variations of these are good approximations to low-energy behaviour. Here we shall consider weakly non-linear $d$-dimensional systems with a periodic potential [8] and magnetic field described by

$$i\partial_z \Psi(\vec{x}, z) = [(-i\nabla - \vec{A}(\vec{x}))^2 + V(\vec{x}) - f(\vec{x}, |\Psi|^2)]\Psi(\vec{x}, z), \tag{1}$$

where $z$ can be either time (BEC) or the distance along the direction of propagation (Optics, here $z \geq 0$). We also assume that $\vec{A}(\vec{x} + \vec{a}) = \vec{A}(\vec{x}); V(\vec{x} + \vec{a}) = V(\vec{x})$ for all $\vec{a} \in \Lambda$, where $\Lambda$ is a $d$-dimensional lattice. Note that in general the assumption is not true for the magnetic potential $\vec{A}$ and we should include disorder [14]. We denote the linear part by $\mathcal{H}_l = (-i\nabla - \vec{A})^2 + V$. We can simultaneously include systems with a boundary in our discussion by splitting $\vec{x} = (x_\perp, \vec{x}_{||})$ and making $V(\vec{x}) = A(\vec{x}) = 0$ for $x_\perp \leq 0$ and periodic only in the $\vec{x}_{||}$-direction with respect to $\Lambda_{||}$, a $d-1$-dimensional lattice, parallel to the boundary. We would like $\mathcal{H}_l$ to correspond to a linear topological system (phase) with a gap [7] or gapped bulk condition (systems with boundary) [15] under adiabatic evolution [16] at a fixed energy scale $\Delta_{gap}$ centered around the *level* $E_{gap}$, as there is no analogue of the Fermi energy $E_F$ for our non-linear systems, since they are generally bosonic [1], [17].

   For fully periodic systems with a gap, there is a coarse classification [1] where a linear topological phase is an element $[\mathcal{H}_l - E_{gap}I] \in \tilde{K}^0(\mathbb{T}^d)$, the $K$-theory group constructed out of vector bundles over the Brillouin torus $\mathbb{T}^d$ arising from the Bloch bands below the Fermi energy [18], [19]. Meanwhile, for systems with a boundary, $K$-theory arises naturally and $[\mathcal{H}_l - E_{gap}I] \in K^{-1}(\mathbb{T}^{d-1})$, where $\mathbb{T}^{d-1}$ is the surface Brillouin torus [15]. Ideally we would imitate the linear classification for systems with $f(\vec{x}, |\Psi|^2)$ by defining a non-linear gap condition and a notion of non-linear adiabatic evolution [20]. The problem is that so far there is no analogue of a gap condition for non-linear systems [9]. We will consider the simplified problem of classifying the topological behaviour of modes (linear perturbations) around soliton solutions, stationary solutions of the form $\Psi(\vec{x}, z) = e^{-i\lambda z}\Phi^\lambda(\vec{x})$ to eq. (1), which decay exponentially as we go to spatial infinity. Have we lost all the interesting non-linear properties by linearizing a non-linear problem around a particular solution? Not at all! In fact, it is the main tool we have to study non-linear stability and existence properties of solitons [8], [13], [21].

   But, can all solitons have topological modes? And even if some do, could these be destroyed by the non-linearity? If a soliton is unstable it will eventually disappear and its modes together with it. Positive (ground state) solitons have two types of instabilities, one

---

[1]Coarse means we do not care about adding trivial valence bands to our systems.

is a *focusing* instability [8], where without increasing, its energy focuses towards a single point, yielding an arbitrary high density which blows up. The other is a *drift* instability, where, via asymmetric distortions, infinitesimal displacements of its original position make it drift towards infinity [8]; therefore, we shall further impose some stability conditions [8]. However, how compatible are these conditions with those necessary for topological modes? Do these conditions have a topological interpretation? We shall elucidate their topological character and their relation to the other topological restrictions in what follows.

## 2  Linearization and stability conditions

We now consider linear perturbations to a given soliton $\Phi^\lambda$ i.e. solutions of the form $\Psi = \Phi^\lambda + \chi_1 + i\chi_2$, such that $\vec{\chi} = (\chi_1, \chi_2)$ satisfies the following linearization of eq. (1) around $\Phi^\lambda$ using a Frechet derivative [8], [21]

$$\partial_z \vec{\chi} = \mathcal{L}(\lambda)\vec{\chi}, \tag{2}$$

where

$$\mathcal{L}(\lambda) = \begin{pmatrix} 0 & L_-(\lambda) \\ -L_+(\lambda) & 0 \end{pmatrix}, \tag{3}$$

with self-adjoint operators

$$\begin{aligned} L_-(\lambda) &= \mathcal{H}_l - \lambda I - f(x, |\Phi^\lambda|^2), \\ L_+(\lambda) &= \mathcal{H}_l - \lambda I - f(x, |\Phi^\lambda|^2) - df|_{\phi^\lambda}. \end{aligned} \tag{4}$$

The linear perturbation is called a *mode* of $\Phi^\lambda$. Have we not lost all the interesting information about our non-linear system by considering the linearized problem? This is in fact one of the main methods for studying non-linear properties, that is, via linearizing around a soliton and using the properties of the linearized operators $L_+(\lambda), L_-(\lambda)$ together with iterative methods to determine properties such as the non-linear stability of a given soliton or existence of other soliton solutions [13], [21], [22]. Our linearized problem has a clear analogue of a gap condition for the mode operators $L_\pm(\lambda)$. We note that $\mathcal{H}_l - \lambda I$ has its spectrum shifted and hence the level at which the soliton satisfies a *gapped modes* condition is at

$$E_{modes}(\lambda) = E_{gap} - \lambda; \; 0 < \lambda < E_{gap}. \tag{5}$$

Thus, the first constraint we put on solitons so that modes can satisfy a gapped modes condition is $\lambda < E_{gap}$. We also have extra potentials determined by $V^1_{\Phi^\lambda} \equiv f(x, |\Phi^\lambda|^2)$ and $V^2_{\Phi^\lambda} \equiv f(x, |\Phi^\lambda|^2) - df|_{\phi^\lambda}$, which we name the *soliton potentials*. For the soliton potentials not to destroy the gapped modes condition, we need them to behave as a perturbation/impurity/defect, in other words, to not fill in the gap with the extra potential, we need

$$\sigma_c(\mathcal{H} - \lambda I + V^1_{\Phi^\lambda} - E_{modes}(\lambda)) = \sigma_c(\mathcal{H} - E_{gap}), \tag{6}$$

$$\sigma_c(\mathcal{H} - \lambda I + V^2_{\Phi^\lambda} - E_{modes}(\lambda)) = \sigma_c(\mathcal{H} - E_{gap}) \tag{7}$$

where $\sigma_c$ denotes the continuous spectrum associated to scattering states. Now the multiplication operator $\Phi^\lambda$ by the very definition of being a soliton, eventually decays exponentially. There is a well known theorem due to Weyl which says that such terms $V^i_{\Phi^\lambda}$ only change the pure point spectrum and do not affect $\sigma_c$ [8], [13], [23]. Thus, the soliton potentials will in general include defect modes, but will not break the gapped modes condition for most non-linearities considered. Defect modes created by the $V^i_{\Phi^\lambda}$'s will play an essential role in section 5.

The above discussion had a caveat and here arises our first connection with instabilities. Positive (ground-state) solitons that are focusing unstable [8], that is, solitons whose modulus blows up become unbouded and hence can modify the continuous spectrum [23]. This means they may eventually break the gapped modes condition, implying that our solitons should satisfy the Vakhitov-Kolokolov stability condition

$$\frac{dP}{d\lambda} < 0, \tag{8}$$

where $P = \int |\Psi|^2 d\vec{x}$ is the particle number (BEC) or optical power, which is conserved.

Let us now consider solitons that are drift stable, i.e. those which stay put under small displacements of their initial position. These have to satisfy the spectral condition $n_-(L_+(\lambda)) = 1$, where $n_-$ denotes the number of negative eigenvalues. Note that for positive solitons, these conditions are necessary and sufficient for full stability [8]. The latter condition can be interpreted as topological if we further note that from the positivity of $\mathcal{H}_l$, there is a restriction on the continuous spectrum $\sigma_c(L_+(\lambda))$ (scattering modes) to be positive. The component of our modes associated to $L_+(\lambda)$ lives in a Hilbert space $\mathfrak{H}^2(\mathbb{R}^d, \mathbb{C})$ and the above means there is a natural split

$$\mathfrak{H}^2(\mathbb{R}^d, \mathbb{C}) = \mathfrak{H}_{-1}(\lambda) \oplus \mathfrak{H}_{\geq 0}(\lambda). \tag{9}$$

The set of all such 1-dimensional subspaces $\mathfrak{H}_{-1}(\lambda)$ of $\mathfrak{H}^2(\mathbb{R}^d, \mathbb{C})$ forms a topological space known as the infinite dimensional Grassmannian $Gr_1(\mathfrak{H}^2(\mathbb{R}^d, \mathbb{C}))$ [24], which we denote $Gr_1$ for shortness. The space $Gr_1$ is a classifying space for the second cohomology group $H^2$ [24], [25]. This means that (up to homotopy) maps from any space $X$ to $Gr_1$ are used to construct the cohomology group $H^2(X; \mathbb{Z})$. Note that the gapped modes condition never entered into our discussion of drift stability. These two conditions are independent as the drift stability is about what happens below $\sigma_c(L_+(\lambda))$, while the gapped modes condition is about what happens in between (same for gapped Bulk-modes). Thus, we can view linearization around the soliton as a map I, such that

$$\Phi^\lambda \mapsto \left\{ \begin{pmatrix} 0 & P_{\geq 0}(\lambda)L_-(\lambda) \\ -P_{\geq 0}(\lambda)L_+(\lambda) & 0 \end{pmatrix}, P_{-1}(\lambda) \right\}, \tag{10}$$

where $P_{-1}(\lambda)$ and $P_{\geq 0}(\lambda)$ are projections to $\mathfrak{H}_{-1}(\lambda)$ and $\mathfrak{H}_{\geq 0}(\lambda)$. Because of conditions (6), (7), the first component can be seen to be equivalent, up to mode adiabatic evolution (for definition see sec (3)), to a Hamiltonian operator in $Gap(L^2(\mathbb{R}^d), \mathbb{Z}^d)$, the space of gapped $d$-dimensional single-particle $\mathbb{Z}^d$-periodic Hamiltonians. In [19], using Bloch's theorem, it is shown using the periodicity that $Gap(L^2(\mathbb{R}^d), \mathbb{Z}^d)$ is coarsely equivalent (by adding trivial bands) to $Map(\mathbb{T}^d, BGL_\infty)$, the space of continuous maps from the $d$-dimensional Brillouin torus to the classifying space $BGL_\infty$, which can be thought of as an ever increasing sequence of Grassmannians. Maps (up to homotopy) to $BGL_\infty$ give rise to the group $\tilde{K}^0$ [26]. Analogously, for systems with a boundary, we instead have $Gap_{Bulk}(L^2(\mathbb{R}^d), \mathbb{Z}^{d-1})$, itself being equivalent to $Map(\mathbb{T}^{d-1}, \mathcal{F}_*^{sa}(\mathfrak{H}))$ [15], where $\mathbb{T}^{d-1}$ is now the surface Brillouin torus and $\mathcal{F}_*^{sa}(\mathfrak{H})$ is a subspace of self-adjoint Fredholm operators [27]. Maps to $\mathcal{F}_*^{sa}(\mathfrak{H})$ now give rise to the group $K^{-1}$ instead of $\tilde{K}^0$. The projection $P_{-1}(\lambda)$ in the second component represents a point in $Gr_1$, as discussed previously.

## 3 Mode adiabatic evolution

Consider now eq. (1) with potential, gap energy and non-linearity, which are also $z$-dependent but in such a way that the linearized evolution of the modes around the soliton

is adiabatic [16], which is guaranteed if the linearized operators $L_{\pm}(\lambda, z)$ satisfy a gapped modes condition for all values of $z$. We set $\Delta_{gap}$ as the adiabatic scale given by the size of the gap and set $s = z/\Delta_{gap}$ to be the dimensionless variable that replaces $z$. Two solitons $\Phi^{\lambda_0}(V_0, f_0)$, $\Phi^{\lambda_1}(V_1, f_1)$ have adiabatically equivalent modes if the modes of one can be evolved into the modes of the other via becoming modes of the solitons $\Phi^{\lambda(s)}(s)$ belonging to a parameter-dependent family of systems $f(s), V(s)$. We name this key concept *mode adiabatic evolution*. We now use the homotopy interpretation [7], [19] for our mode adiabatic evolution. Two different solitons $\Phi^{\lambda_0}(V_0, f_0)$, $\Phi^{\lambda_1}(V_1, f_1)$ have modes in the same homotopy class if there exists an $s$-dependent family of soliton solutions $\Phi^{\lambda(s)}(V(s), f(s))$ such that there is a continous family of maps $I(s)$ from eq. (10), which restricts to the maps $I_0$ (coming from $\Phi^{\lambda_0}(V_0, f_0)$) at zero and $I_1$ (coming from $\Phi^{\lambda_1}(V_1, f_1)$) at one respectively i.e. a continuous path in $Map(\mathbb{T}^d, BGL_\infty) \times Gr_1$, which describes their evolution.

Thus, using this interpretation, we can separate modes into topological classes and so, we can separate individual solitons too, according to the corresponding topological character of their modes. Employing the homotopy type of the spaces discussed above, we have that for periodic systems the set of distinct classes of topological modes around solitons is equivalent to the groups:

$$\tilde{K}^0(\mathbb{T}^d) \oplus H^2(*; \mathbb{Z}), \tag{11}$$

where $*$ denotes (from here on) a point, viewed as a topological space. For systems with a boundary we replace $Gap(L^2(\mathbb{R}^d), \mathbb{Z}^d)$ with $Gap_{Bulk}(L^2(\mathbb{R}^d), \mathbb{Z}^{d-1})$ and using the results of [15], we obtain $K^{-1}(\mathbb{T}^{d-1}) \oplus H^2(*; \mathbb{Z})$ instead. We remark that many solitons of interest, such as those that are surface-localized, are often gap solitons [2], [28] and do not satisfy the spectral condition mentioned above. The topological interpretation of the drift stability condition might seem irrelevant since the group $H^2(*; \mathbb{Z})$ is trivial, but we shall see it yields new classes for systems with more symmetry.

# 4 Crystallographic symmetries

Consider systems which further have a crystallographic symmetry with point group $P \subset O(d)$ [29], [19]. If we restrict to $P$-symmetric soliton solutions, their corresponding $L_{\pm}(\lambda)$ will be $P$-invariant. Further, if they satisfy all of the conditions discussed above and the mode adiabatic evolution respects this $P$-invariance, then the crystalline topological classes of modes around $P$-symmetric positive solitons are given by:

$$\bar{K}_P^{0,\tau}(\mathbb{T}^d) \oplus H_P^2(*; \mathbb{Z}). \tag{12}$$

The groups $\bar{K}_P^0$ and $H_P^2$ denote a twisted equivariant version of $K$-theory [2], [19] and equivariant cohomology [30], [31], respectively. The interesting thing here is that $H_P^2(*; \mathbb{Z})$ is no longer trivial! Instead it is equivalent to $H^2(BP; \mathbb{Z})$, where $BP$ is an infinite dimensional space known as the classifying space of $P$ [25]. To have an example in mind note that for $P = \mathbb{Z}_2$, $B\mathbb{Z}_2 \simeq \mathbb{R}P^\infty$, the infinite dimensional real projective space. Hence, the spectral condition for drift stability (9) becomes topologically non-trivial when we include more symmetries. For systems with a boundary we replace $\bar{K}_P^{0,\tau}(\mathbb{T}^d)$ with $K_P^{-1,\tau}(\mathbb{T}^{d-1})$, where $P$ now denotes surface crystallographic symmetry and $d \geq 2$ [32].

What is the physical interpretation of these classes? On the one hand for positive solitons, $\mathfrak{H}_{-1}$ is a direction of instability which has to be controlled [13]. For the perturbed

---

[2]By $\bar{K}_P^0(\mathbb{T}^d)$ we mean the kernel $K_P^0(\mathbb{T}^d) \to K^0(*)$. This is because the coarser classification does not care about adding trivial bands and changing the dimension of our bundles.

| $d$ | Boundary (y/n) | $P$ | L+DS |
|---|---|---|---|
| 2 | n | 0 | $\mathbb{Z}$ |
| 2 | y | 0 | $\mathbb{Z}$ |
| 2 | n | pm | $\mathbb{Z}^2 \oplus \mathbb{Z}_2$ |
| 3 | y | pm | $\mathbb{Z}^3 \oplus \mathbb{Z}_2$ |

Table 1: Four examples of topological classes for modes around solitons in $d$-dimensional systems, where $P$ is the crystallographic point group, L+DS stands for linear plus drift stable topological classes

solution to remain at most $\epsilon$-distance from $\Phi^\lambda$ at any value of $z$, the initial perturbation needs to be at a distance $\delta(\epsilon, \mathfrak{H}_{-1})$ from $\Phi^\lambda$. As we adiabatically evolve $\mathcal{L}(\lambda, s)$, we would expect an $s$-dependence $\delta(\epsilon, s)$; however, the topological character of $\mathfrak{H}_{-1}$ and $P$-symmetry will mean that $\delta$ is $s$-independent. Thus, how close the initial perturbation has to be to our soliton depends only on the topological action of $P$ on $\mathfrak{H}_{-1}$ and the $\epsilon$ chosen. We remark that this new invariant does not arise in linear systems.

We present a few examples in dimension $d = 2, 3$ with and without $pm$-symmetry in Table 1.

## 5   Spaces of soliton solutions, global classes and bifurcations

So far our analysis tells us the different topological character of individual solitons, but does a single soliton define the character of eq. (1)? Given $\vec{A}, V$ and $f$ there will generally be many solitons which satisfy conditions (5, 6, 7, 8). Let us first, for simplicity, consider $\vec{A} = 0$, $V = 0$, $f = |\Psi|^{2\sigma}$, the most basic type of NLS equation. This equation has a ground state soliton $\Phi^0(\vec{x}) > 0$ and it is easy to see that if $\Phi^0$ is a solution so is $\Phi(\theta, \vec{x}_0, \vec{v}, R) = e^{i\theta}\Phi^0(R(\vec{x} + \vec{x}_0) + \vec{v}t)$ with $(R, \vec{x}_0, \vec{v}) \in E(d) \times \mathbb{R}^d$ i.e. eq. (1) has as its symmetry group the Euclidean group $E(d)$ times Galilean boosts $\mathbb{R}^d$ times a phase $S^1$, because $|e^{i\theta}\Psi|^{2\sigma} = |\Psi|^{2\sigma}$. The group action (the symmetries) generates a set of solutions

$$\mathcal{M} = E(d) \times \mathbb{R}^d \times S^1. \tag{13}$$

The space $\mathcal{M}$ is known as the *Soliton manifold* [21], [22], [33]. The soliton manifold is extremely relevant to the non-linear dynamics, as part of said dynamics is described as motion along $\mathcal{M}$, which is called a modulation equation. However, from an algebraic topology point of view, this symmetry is a redundancy (analogous to gauge symmetry) since we could obtain all the equivariant information by the action of $E(d) \times \mathbb{R}^d \times S^1$ on $\Phi^0$. Hence, we further identify all points in $\mathcal{M}$ which are simply symmetry translates of the ground state. Abusing notation we will denote this quotient space $\mathcal{M}$, which in this particular case, i.e. with no potential, is a point $* = E(d) \times \mathbb{R}^d \times S^1 / E(d) \times \mathbb{R}^d \times S^1$.

### 5.1   Symmetry breaking and non-trivial soliton manifold

Let us now include a potential $V$ and let us call the symmetry group of $V$, $G_{symm} \subset E(d)$. Similarly as above, $G_{symm}$ generates a soliton manifold [33]. It is well known [34] that even for $f(|\Psi|) = -g|\Psi|^2$ there is a critical value $g^*$ such that for every $g \geq g^*$ there is *spontaneous symmetry breaking* of the ground state into a lower dimensional symmetry group, which we denote $G_{Sol} \subset G_{symm}$. In precisely the same fashion as for lattice

defects [35] the stable ground state soliton manifold becomes [36]

$$\mathcal{M} = G_{symm}/G_{Sol}. \tag{14}$$

In general this coset space has a non-trivial homotopy type. We shall see how this non-trivial topology gives rise to different purely *non-linear* topological phases.

## 5.2  Non-linear topological phases and K-theory

The soliton manifold $\mathcal{M}$ is a property of the non-linear system in (1) which does not arise in linear systems. As we have stated before to study the implications of $\mathcal{M}$ on the dynamics, essentially the main tool to do so is to once again linearize around a soliton $\Phi^\lambda \in \mathcal{M}$. We can thus consider the linearization $\mathcal{L}_{\vec{A},V,f}$ in (4) with $\lambda = 0$ as a family of operators parametrized by $\mathcal{M}$ into some space of operators.

$$\mathcal{L}_{\vec{A},V,f} : \mathcal{M} \longrightarrow \mathcal{F}^A(\mathscr{H}). \tag{15}$$

Then, if we mode adiabatically evolve the system $(\vec{A}(s), V(s), f(s))$, $\mathcal{M}(s)$ will also change and not necessarily in a continuous fashion. However, as long as $\mathcal{M}(s)$ is a homotopy of spaces, we can use $\mathcal{M} = \mathcal{M}(0)$ as a fixed characteristic of our phases and study the family $\mathcal{L}_{\vec{A},V,f}$ up to homotopy to define global topological classes.

What do we know about the space of operators $\mathcal{F}^A(\mathscr{H})$? From equation (4) setting $\lambda = 0$ we know that the self-adjoint components $L_+, L_-$ have finite dimensional kernel and hence so does $\mathcal{L}_{\vec{A},V,f}$ and its adjoint $\mathcal{L}^\dagger_{\vec{A},V,f}$. Thus $\mathcal{L}_{\vec{A},V,f}$ is a Fredholm operator [27]. Note that we may extend to $\lambda \neq 0$ as long as $\mathcal{L}_{\vec{A},V,f}(\lambda)$ is still Fredholm, which will happen for some open interval around $\lambda = 0$. There is one final property of $\mathcal{L}_{\vec{A},V,f}$ we have to consider, which is a stability requirement on its spectrum at each point in $\mathcal{M}$ satisfies [13]

$$\sigma(\mathcal{L}_{\vec{A},V,f}(\Phi)) \subset i\mathbb{R} \tag{16}$$

It is not difficult to show that the above conditions on the spectrum, Fredholm and self-adjoint character of its components imply that $\mathcal{F}^A(\mathscr{H}) \simeq \mathcal{F}^{sa}_*(\mathscr{H})$, the $*$-component of the space of self-adjoint Fredolm operators [27] that we had employed in section 3 for gap modes on systems with boundary [15]. Thus, using mode-adiabatic evolution as homotopy classes of maps, we can construct an invariant which distinguishes non-linear topological phases

$$[\mathcal{M}, \mathcal{F}^{sa}_*(\mathscr{H})] = K^{-1}(\mathcal{M}). \tag{17}$$

This invariant will most likely not be sufficient for a complete classification of non-linear topological phases since, for a given system, there are other soliton manifolds (for larger number of solitons) [37] which we will consider in future work. Nevertheless, this is the first step towards a systematic classification of non-linear topological phases.

### 5.2.1  Extension to topological solitons

We can consider extensions of the NLS/GP equation to equations with internal degrees of freedom and/or a dynamical gauge field such as in the case of the $U(1)$-Landau-Ginzburg equation [38] or spinorial versions, including a nonlinear Dirac equation [39]. In all of these generalizations there is also spontaneous symmetry breaking giving rise to topological solitons [40], with their soliton manifold also given by eq. (14). However, in those cases $G_{symm}$ is an *internal* symmetry. One can once again consider linearization around these solitons (for example Majorana zero modes on vortices) and repeat all of our entire analysis. We expect similar results but we shall postpone our analysis for future work.

## 5.3 Geometric interpretation as global bifurcations and computations

Now that we have constructed a partial classification for non-linear topological phases, do we have any kind of interpretation for these? Since the mid 1970's to date there have been many applications of algebraic topology to the study of global bifurcations in non-linear PDE systems e.g. [41], [42], [43] to mention a few of the original references. In particular the work of Pejsachowicz [43], [44], [45] (see [46] for a review and some improvements) is most relevant for our purposes. How does Pejsachowicz study bifurcations? They consider nonlinear Fredholm maps from a multiparameter space $X$ cross a Hilbert space to the same Hilbert space i.e. $R : X \times \mathscr{H} \longrightarrow \mathscr{H}$ and linearize by taking the Frechet derivative of $R$ at points $(x, 0) \in X \times \mathscr{H}$ that is $DR_x(0)$. They then consider the index bundle associated to this linearization $[DR] \in KO(X)$ and show that it counts the number of bifurcation points from the trivial branch at $X \times \{0\}$. Not only does a non-trivial index bundle signal bifurcations, these are *global* (see chapter 9 of [47] for a pedagogical introduction) in the sense that they carry information beyond a small neighborhood of the bifurcation point and may also signal bifurcations of orbits e.g. hetero and homoclinic orbits, periodic orbits, stable and unstable manifolds etc [48].

In our case the role of the multiparameter space is played by the soliton manifold $\mathcal{M}$ and the function $R$ is given by moving everything in (1) to one side and evaluating at $\Phi \in \mathcal{M}$. We should clarify that in our case $DR_x(0) \in \mathcal{F}^A(\mathscr{H})$ instead of Fredholm operators in a real Hilbert space as in [44], [46]. Do solitons bifurcate from $\mathcal{M}$? This is in fact the bread and butter of results in the study of the NLS/GP equation [21], [22] including bifurcations at symmetry breaking [34]. However the study of these bifurcations is usually either of a local nature i.e. using the Lyapunov-Schmidt method or global, but studied using singularity theory [33]. We leave for future work the exact implications for bifurcations of having a non-trivial element in $K^{-1}(\mathcal{M})$ as different types of topological invariants signal different phenomena in global bifurcation theory e.g the spectral flow and periodic orbits in Hamiltonian systems [45] or trajectories connecting bifurcation points [44].

### 5.3.1 Qualitative example

Consider eq. (1) with a spherically symmetric potential i.e. $V = V(|\vec{x}|)$ in $d = 2$. The group of symmetries is $G_{symm} = SO(2)$, however, it can be spontaneously broken into $G_{Sol} = \mathbb{Z}_2$ [36]. Thus,

$$\mathcal{M} = SO(2)/\mathbb{Z}_2 \simeq S^1. \tag{18}$$

Our construction predicts

$$K^{-1}(S^1) = \mathbb{Z}. \tag{19}$$

This integer invariant is called the spectral flow [49] and we can interpret as resonances (defect modes) bifurcating from the scattering modes (radiation) of negative frequency to positive frequency as depicted in Figure 1.

Currently technical details prevent us from explicitly matching this result with those of [45]. Nevertheless, we conjecture that we can indeed match them and that nonlinear systems with a nonzero spectral flow, as depicted in Figure 1, have a global bifurcation of their periodic orbits, where as those which have zero spectral flow do not.

We also note that for the much less studied case of a non-linearity $f$ that does depend on the phase i.e. $f(\Psi) \neq f(|\Psi|)$ like the ones considered in [50], we now have (without spontaneous symmetry breaking), after symmetry identifications, that the soliton manifold is also a circle and hence admits the same interpretation as depicted in Figure 1 except that now the $\theta$-axis denotes the phase dependence of our solitons.

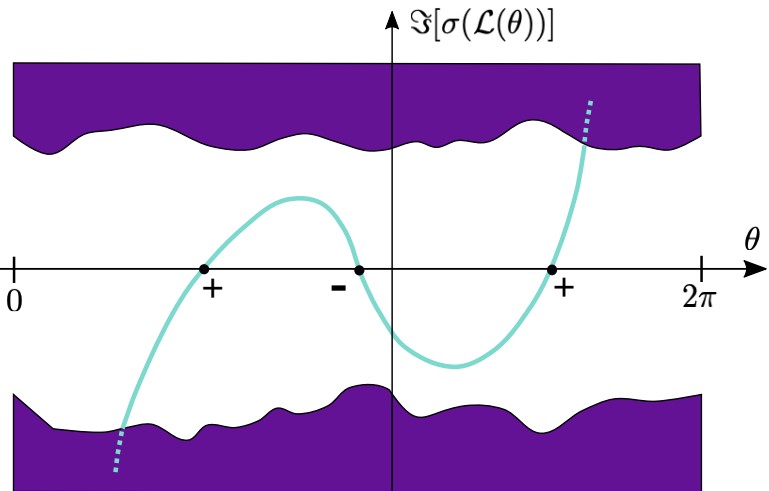

Figure 1: $d = 2$, $G_{symm} = SO(2)$, $G_{Sol} = \mathbb{Z}_2$. The $\theta$-axis represents the phase parametrizing a family of soliton solutions which have spontanously broken $SO(2)$-symmetry into $\mathbb{Z}_2$. $\Im[\sigma(\mathcal{L}(\theta))]$ denotes the imaginary part of the spectrum of $\mathcal{L}_{(\vec{A},V,f)}(\theta)$. The solid purple regions represent the continuous spectrum associated to scattering states (radiation). Dashed blue lines entering the continuous spectrum are defect modes embedded into scattering states. Blue solid lines represent bounded defect modes arising between the gap. The $\pm$ signs denote the sign of the slope as defect modes cross the $\theta$-axis. The spectral flow depicted here is equal to $2(+) - 1(-) = 1$. Conjecturally, a nonzero value signals a global bifurcation of the periodic orbits of the system as in [45]

## 5.4 Comments about the physical interpretation

We have presented a generic geometric interpretation of $K^{-1}(\mathcal{M})$ as signalling a global bifurcation (as in the above example) but what about the physical implications? In linear topological phases the topological invariants like the Chern number appear in the formula for some macroscopic property of the system like conductivity. Beyond the physical interpretation that distinct elements of $K^{-1}(\mathcal{M})$ represent distinct non-linear topological phases, what can we say about macroscopic properties of the phases they represent? On the one hand we can clearly interpret, as in the example depicted in Figure 1, that these non-linear topological phases have to do with the behaviour of defect modes and resonances of scattering states as we move in the soliton manifold $\mathcal{M}$, generated by spontaneous symmetry breaking. In the linear case, the topological character of zero modes of individual defects, generated at spontaneous symmetry breaking, has been studied in [51], [52] and [53]. In its simplest form they consider a sphere $S^D$ surrounding the individual soliton in real space (the soliton which was created at symmetry breaking) and give an interpretation of the conductance and polarization contributed by the defect in terms of topological invariants of $S^D$ like $K^0(S^D)$ and its many variants when we include symmetries. Thus, one might suspect that there is some relation as we are also building our invariants from the behaviour of the defect modes of solitons arising from symmetry breaking, however, we are using the soliton manifold $\mathcal{M}$ instead of their sphere $\mathcal{S}^D$ so the relation is not so direct. Atiyah and Singer [54] considered the Dirac operator coupled to a background instanton (analogous to our linearization around a soliton) and show that the $K$-theory of the instanton moduli space $\mathscr{A}$ (analogous to our soliton manifold $\mathcal{M}$) i.e. $K^0(\mathscr{A})$ has a physical interpretation in terms of an anomaly. So it seems reasonable that our non-trivial non-linear topological phases represent an anomaly suitably interpreted in the optical and exciton-polariton case. We leave the precise connection for future work. On the other hand, we expect that the geometric interpretation in terms of a global bifurcation translates directly into a macroscopic property. An example of this type of relation was found in ref. [55], where a new, purely nonlinear type of power-oscillations arise when there is a Hopf bifurcation. We also note that the topological bifurcation methods discussed in subsection 5.3 can detect global Hopf bifurcations [56]. The system considered in [55] seems like a strong candidate to be in a nontrivial non-linear topological phase but since it requires considering complex terms in eq. (1) and including symmetries such as chiral or charge conjugation, we will not address it here.

## 6 Conclusions

We have attacked, for the first time (to our knowledge), the problem of systematically assigning topological invariants to non-linear topological systems in photonics, exciton-polaritons and BECs, simultaneously. Our analysis provides the conditions for which the modes around solitons have the same topological character as linear phases do and describes how soliton stability conditions become topologically non-trivial when including crystallographic symmetries. These crystalline stability conditions yield new invariants for these modes, which have no analogues in linear systems. Using the space of soliton solutions (soliton manifold) which are symmetry breaking, we built novel, global, purely non-linear invariants in terms of the $K$-theory of the soliton manifold, providing a partial answer to the problem of classifying non-linear topological systems. We further provide a qualitative example for $d = 2$ and give some evidence towards our conjecture that these non-linear topological phases signal a global bifurcation for the systems in question. We also briefly discuss how similar global invariants must exists for generalizations with topo-

logical solitons and further provide some connection from the existing literature between physical properties of the system, soliton defect modes and a global bifurcation.

Our analysis employed a linearization around a soliton solution which is the standard way of studying non-linear systems and which, as discussed in section 5, can yield information about global bifurcations. General optical systems are non-hermitian as they have gains and losses, thus the natural mathematical paradigm to provide a comprehensive classification is that of non-hermitian systems [57]. Hence, the next step in the extension of this work is to consider the $K$-theory arising from the point and line gap generalizations for these systems but also to include exceptional points [58]. We also have not included in our discussion the generalizations of particle-hole and time reversal symmetries appearing in the periodic table of linear topological phases [18], [19]. Finally we could consider extending all of the above to so called gap solitons [28], [10], [22] which bifurcate from the spectral edges of the linear problem however much less is known about their stability requirements and about their soliton manifold.

# Acknowledgments

We especially thank Professor S. Gustafson for explaining various facts about solitons. We also thank Professors O. Antolín-Camarena, K. Ramos-Musalem and J. Sheinbaum for useful comments. Finally we acknowledge funding from CONACYT Frontera 42821.

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
