# Peer review of "Solitons in Weakly Non-linear Topological Systems: Linearization, Equivariant Cohomology and K-theory"

_SciPost Physics_

## Round 1 · Referee Report · Anonymous (Referee 1) · 2023-3-27

Report

This manuscript is a proposal to extend the notions of topological insulators to a nonlinear framework. This is an ambitious and open question. The main idea is to investigate the topological properties of linear perturbations around a soliton solution of the NLS equation. Independently, the manuscript also considers the topological aspects of the space of nonlinear solitons solutions directly. The study relies on Fredholm theory, K-theory and equivariant cohomology.

These ideas seem fairly reasonable and lots of statement are made along the manuscript, but those remain very broad and are not supported by any proof or explicit example. Advanced mathematical concepts are used to tackle the problems, but the manuscripts constantly jumps to a new concept and leaves several unsolved issues without answer. Finally, the physical relevance of the discussion is questionable : the manuscript lacks of a specific and accurate model for which all the hypothesis made along are satisfied. It is actually not clear if such a model exists. Such a problem is actually already acknowledged here and there in the manuscript.

For the linearized part (section 2, 3 and 4), it is not clear that there exists a soliton solution to NLS equation when the linear part is topological (in particular with a non-periodic vector potential), and if so, that it is stable and satisfies (8). What would be the topological index associated to such a solution? Moreover, section 4 involves a crystallographic symmetry, which also questions the existence of NLS-solitons in presence of such a symmetry. Finally, the end of section 3 involves Bloch theorem and Brillouin torus, whereas the soliton potential appearing in (4) is never periodic.

The global classification from Section 5 seems completely orthogonal to the rest. Figure 1 looks indeed very much like edge modes of (linear) topological insulators, but again the whole discussion should be supported by an explicit model or class of models, rather than a "qualitative example". Also, the physical relevance of such topological properties is not clear, which makes the analogy with topological insulators very distant and does not really answer the original question.

For all these reasons, I do not recommend this manuscript for publication in SciPost Physics, unless some drastic changes are implemented in each section of it.

---

## Round 1 · Referee Report · Anonymous (Referee 2) · 2023-5-9

Report

The manuscript studied topological property of non-linear systems. Here are a few questions:

  1. Can the author clarify what is the topological property discussed in the manuscript? Are there excitations with nontrivial braiding or fusion? Are there protected anomalous boundary modes similar to topological insulator? Are there operators that are invariant under smooth deformation of their support?

  2. In Section 5.1 the manuscript discussed topological property for sigma model from symmetry breaking. For instance, if breaking SU(2) symmetry to U(1), there will be M=SU(2)/U(1)=S^2 sigma model. What is the topological property?

---

## Editorial Decision

awaiting_resubmission